# Detection and Isolation of DoS and Integrity Cyber Attacks in Cyber-Physical Systems with a Neural Network-Based Architecture



**Carlos M. Paredes [1], Diego Martínez-Castro [1], Vrani Ibarra-Junquera [2] and Apolinar González-Potes [2,3,*]**

[1] Departamento de Automática y Electrónica, Universidad Autónoma de Occidente, Cali 760030, Colombia; cmparedes@uao.edu.co (C.M.P.); dmartinez@uao.edu.co (D.M.-C.)
[2] Laboratorio de Agrobiotecnología, Universidad de Colima, Colima 28400, Mexico; vij@ucol.mx
[3] Facultad de Ingeniería Mecánica y Eléctrica, Universidad de Colima, Colima 28400, Mexico
[*] Correspondence: apogon@ucol.mx

**Abstract:** New applications of industrial automation request great flexibility in the systems, supported by the increase in the interconnection between its components, allowing access to all the information of the system and its reconfiguration based on the changes that occur during its operations, with the purpose of reaching optimum points of operation. These aspects promote the Smart Factory paradigm, integrating physical and digital systems to create smarts products and processes capable of transforming conventional value chains, forming the Cyber-Physical Systems (CPSs). This flexibility opens a large gap that affects the security of control systems since the new communication links can be used by people to generate attacks that produce risk in these applications. This is a recent problem in the control systems, which originally were centralized and later were implemented as interconnected systems through isolated networks. To protect these systems, strategies that have presented acceptable results in other environments, such as office environments, have been chosen. However, the characteristics of these applications are not the same, and the results achieved are not as expected. This problem has motivated several efforts in order to contribute from different approaches to increase the security of control systems. Based on the above, this work proposes an architecture based on artificial neural networks for detection and isolation of cyber attacks Denial of Service (DoS) and integrity in CPS. Simulation results of two test benches, the Secure Water Treatment (SWaT) dataset, and a tanks system, show the effectiveness of the proposal. Regarding the SWaT dataset, the scores obtained from the recall and F1 score metrics was 0.95 and was higher than other reported works, while, in terms of precision and accuracy, it obtained a score of 0.95 which is close to other proposed methods. With respect to the interconnected tank system, scores of 0.96, 0.83, 0.81, and 0.83 were obtained for the accuracy, precision, F1 score, and recall metrics, respectively. The high true negatives rate in both cases is noteworthy. In general terms, the proposal has a high effectiveness in detecting and locating the proposed attacks.

**Keywords:** anomaly detection; anomaly isolation; artificial neural networks; Cyber Physical System

## 1. Introduction

Cyber Physical Systems (CPSs) emerge from the attempts to unify the emerging applications of embedded computers and communication technologies used to monitor, control, as well as generate actions on physical elements to fulfill with a specific task [1], and they have an important impact on different sectors [2].

The different parts of the system are usually interconnected using communication networks to share information and data that interact with each other and, sometimes, cloud computing services [3–5]. CPSs can be represented in layers, as shown in Figure 1. The first is the physical layer, where the physical infrastructure of the system, sensors, and actuators are located, with the objective of monitoring and controlling physical processes.

The second is the network layer, which implements the transmission data and allows the interaction between the physical layer and the cybernetic layer. Finally, a cybernetic layer allows the abstractions of the received data, as well as the interaction between networks, devices, and the physical infrastructure [6].

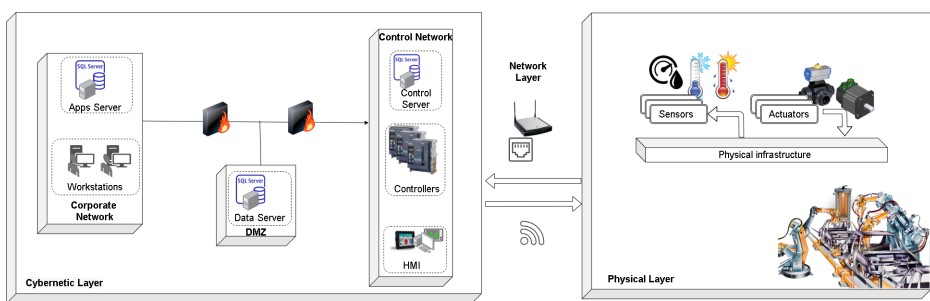

**Figure 1.** Architecture of a CPS.

Society currently relies on multiple automatic systems supported by CPSs. These applications are focused in contexts, such as industrial, health, and environmental, among others [7,8]. Security and reliability are fundamental requirements in these systems. Cyber attacks can generate inappropriate behaviors and catastrophic impacts on the physical world, causing damage to both the system infrastructure and industrial products and even threaten human lives [9]. Examples, such as attacks on smart grids, aviation systems, water plants, chemical plants, and oil and natural gas distribution systems, are becoming increasingly high [10–14]. The above has caused this research area to be active in recent years.

Therefore, there must be mechanisms to detect the occurrence anomalies to avoid exploiting vulnerabilities in the devices connected to the system network. Real-time detection is very important in order to ensure reliability and security in these systems, where sensors are prone to malicious attacks. For this reason, detection systems are often used, such as Intrusion Detection Systems (IDS), which monitor data traffic to identify and protect systems from these eventualities. Based on detailed analysis of network traffic and device usage, IDSs seek to evaluate this information to identify unwanted events. IDSs do this by carrying out three stages: monitoring, analysis, and detection. Monitoring relies on a sensor network or host-based sensors, the analysis stage is based on any method to identify and extract features, and the detection stage relies on anomaly detection [15,16].

Within these can be highlighted: [17] the methods based on traditional information technologies, where network traffic analysis is used to detect anomalies [18–26]; and model-based methods, where detection is performed by comparing the system actual output with an expected value [4,27–31].

According to reference [16,32], host-based IDS methods operate on the data collected from the individual parts of the computer systems and can detect internal changes and determine which processes and/or users are involved in malicious activities, which can be not significant with some devices; thus, this method sometimes fails. Whereas a network-based IDS will detect malicious packets as they enter your network or unusual behavior on your network, such as flooding attacks, more traditional IDS can do it on one channel or across the network. These monitor the entire network traffic to detect known or unknown attacks using techniques based on anomalies, signatures, and specifications [16,33,34]. Hence, IDSs help to avoid critical consequences and assist in making appropriate decisions when system events occur by performing two main tasks: attack detection, which decides whether or not an anomaly has occurred; and attack isolation, which decides which elements of the system are being affected by the unwanted.

In such a way, the purpose of this research is to present the design of an architecture that allows detecting and isolating attacks that may occur between the elements of the physical layer and the controller, generating alerts that allow detection and localization of the origin of the cyber attacks. For this, a new architecture was proposed for the

detection and isolation of attacks using techniques based on artificial intelligence. The proposal integrates two approaches: regression and classification. The first approach allows generating models that describe the behavior of the real process to estimate the outputs by using process input data, obtaining in this way the model to be compared with the real process values in order to detect and isolate the attack. The second approach allows generating detection systems to carry out the detection and isolation of attacks. The proposal was subjected to two test benches, obtaining better results than those reported in previous works. The contributions of this paper are as follows:

- The design of an architecture using one-dimensional convolutional neural networks to detect and isolate cyber-attacks that involve the elements of the physical layer and the controller of a CPS, generating alerts to detect and locate the origin of the cyber attack.
- The architecture proposed is an architecture based on the process information, where the dynamic properties of the process are covered, in order to evaluate the possibility of a cyber attack occurring in different parts of the system, without the need to define a threshold that allows separating normal situations with events where a cyber attack is possibly occurring.
- The design of the architecture allows detecting and locating the occurrence of cyber attacks occurring simultaneously in different parts of the system, even when the attacks are of different types.

The remaining sections are structure as follows. Section 2 presents related works. Section 3 describes the problem statement. In Section 4, the attack detection and isolation method is proposed. Section 5 exposes the results obtained using the method proposed in two test benches. Finally, in the last section, we present the conclusions.

## 2. Related Works

Protection systems in industrial processes have used strategies that have presented good performance in other environments, such as office environments. However, the characteristics of these applications are not the same, so the results obtained are not as expected. This is because the availability of equipment in industrial systems is very high; so, in many cases, a simple solution in corporate environments, such as patching, simply does not work because the machine is not available to shut down until a planned outage. It is also difficult to predict how a newly introduced patch will affect the operation of a control system, especially if the patch is not rigorously tested, increasing the organization's reluctance to act on potential threats. The implementation of security patches can affect application performance and, therefore, the stability, availability, and real-time behavior of machines. Something equivalent occurs with the impact on data traffic through the communications network associated with solutions that evaluate network traffic, which can affect delays in control strategies and, in turn, the performance of control loops [35]. This problem has motivated different projects with the purpose of contributing from different approaches to increase the security of control systems. In this section, the related works are described.

Some ongoing projects to improve security in these systems have included methods to provide aspects, such as data confidentiality and authentication, access control, within the network, and privacy and reliability of applications, as well as the inclusion of security and privacy policies [36]. Even so, CPSs are vulnerable to multiple attacks aimed at disrupting the network and modifying process variables, altering its operation. For this reason, new defense mechanisms designed to detect cyber attacks have been generated. One of the best known mechanisms is IDS. IDS approaches may be classified as signature-based, anomaly-based, or specification-based [33].

The signature-based method only detects records that are inside of a database, and it is highly accurate and effective against known threats, consumes more power, and does not detect new events [33]. The anomaly-based method is efficient in detecting new attacks [16] since it compares the system activities in a moment against an usual behavior profile and generates alerts whenever a threshold defined by the system's normal behavior is cross [34].

However, anything that does not match normal behavior is considered an intrusion, and learning all normal behavior is not an easy task. Therefore, this method generally has high false-positive rates. On the other hand, the methods based on specifications use a set of rules and thresholds that define the expected behavior of the different components of the network. It has the same purpose as anomaly-based methods, with the difference that this method is specified manually by an expert who determines the specifications. Manually defined specifications typically provide low false-positive rates versus anomaly-based detection and do not require training steps because it can be used immediately. However, these methods cannot be adapted to different environments and can be time-consuming to adjust and error-prone [33].

Other authors have developed state observers for detection, such as the Luenberger Observer (LO), while the isolation process is realized by structured residues generated using Unknown Input Observers (UIOs) [37–40]. These methods present drawbacks because the detection of anomalies is realized by a comparison of a fixed threshold defined by a historical data of normal behavior, with the difference between the variables of the actual process and the values generated by an estimated model. Then, it can lead to a considerable rate of false positives and false negatives. The above is because, for the design of the observer banks, the knowledge of the parameters and the dynamics of the system is used, which sometimes can be significantly different of the real system performance. So, both proposals are limited by the knowledge of the process, such as the definition of the threshold, which, in real situations, it may not be easy to model accurately.

In the last few years, data-driven methods have been employed to detect cyber attacks [18–23,25,41]. These methods have presented good performance to find models of processes that even present quite pronounced non-linear dynamics. Machine learning technology is one of the data-driven methods emerging as a method to detect attacks in these systems [23,26,42–50].

Random Forest-based algorithms have been employed recently to detect malicious behavior by using databases; in this case, binary classification is applied to classify whether the content of a packet is malicious or not. This method reduces computational cost but does not guarantee high accuracy [51]. In this way, it is not possible to identify which task transmitted the packet, and it does not allow specifying the type of attack [15,16]. From another point of view, in Reference [52], a scheme was proposed to protect remote patient monitoring systems against DoS attacks. An attack detection model was established by developing mechanical learning using decision trees. The model could help to locate various types of attacks, focusing mainly on flooding attacks, and could be appropriate to devices with limited memory and processing resources, such as sensors and healthcare devices. As future work, they propose the possibility of identifying other types of attacks and even developing a mechanism to block a wide range of attacks.

Other approaches have used different artificial intelligence techniques, such as Support Vector Machines (SVMs), genetic algorithms [32], self-organized networks of ant colonies, and extreme learning machines, which provide models with very high accuracies applied in the context of security in computer networks, and especially in the detection of intrusions. The purpose of these techniques is to achieve better intrusion recognition rates, but it is still noticeable that the false positive rate remains the problem to be approached in all these studies. Although some technique can reduce the false positive rate, it increases the training time and classification, which is a relevant index for real-time detection [53].

In Reference [18], an SVM-based algorithm was used to classify normal and abnormal behavior of data traffic that may be subjected to DoS attacks. This algorithm reaches good attacks predictions rate with less training time. In Reference [19], a method based on Principal Component Analysis (PCA) and SVM to detect DoS attacks was presented. The paper analyzes the effects of DoS attacks in a network using TCP protocol. The PCA algorithm is used in order to filter the interference of the environment to extract the main features effectively and reduce the dimensioning of information without losing information from the original data. The results show that the algorithm has high accuracy and a low

false positive and false negative rate (FPR and FNR). In the same context, an SVM using a radial basis kernel function is proposed in Reference [20] to detect attacks in networked automotive systems. This proposal aims to avoid drawbacks associated with cases in which there is not an events dataset, or it is probably not sufficiently representative because many of the possible situations of a system are unknown. However, these techniques are not suitable for detecting mutations from various attacks.

Advanced techniques, such as Deep Belief Networks (DBN) and Deep Convolutional Neural Networks (Deep CNN) [54,55], are trained to extract low-dimensional features and are used to discriminate usual and hacking packets. In Reference [56], an anomaly detector based on a neural network recurrent Long short-term memory (LSTM) was proposed to detect attacks with low false alarm rates. These methods have had the best response in these environments, although the computational costs sometimes are high [20,55]. Thus, applying machine learning and other artificial intelligent techniques is a challenge because it requires more memory and computational cost that can affect the performance of the system.

In addition, to validate the proposal, two test benches were used. For the selection of these datasets, a search was performed that included keywords, such as security in industrial control systems, detection of faults, anomalies and cyber attacks in control systems, and design of secure CPSs. From this search, we considered the publications that had a publication time of less than 5 years, as well as the number of times that the datasets had been used to evaluate the security on CPSs. We also considered the type of attacks that were implemented, since our approach was to address different types of attacks, including those with the highest frequency and impact on the control systems found in the CPSs (integrity and DoS attacks).

The first one corresponds to the SWaT dataset, which provides real data from a simplified version of a real world water treatment plant. This dataset allows researchers to design and evaluate defense mechanisms for CPSs and contains both network traffic and data concerning the physical properties of the system [57]. On the other hand, there is another test bed which consists of three interconnected tanks [58] that has allowed the validation of different types of detection methods for cyber attacks on CPS. These two test benches have made it possible to validate different proposals focused on techniques that allow us, in one way or another, to analyze the detection of cyber attacks [37,42,59–69] and have made it possible to direct this research to improve the proposed proposals.

Based on this review, Table 1 summarizes each of the related reports to a set of characteristics in order to highlight the issues that need to be addressed to improve the strategies and proposals in the future.

**Table 1.** Summary of related works.

| Reference | Main Domain | Technique | Type of Anomaly | Advantages | Limitations | Evaluation |
|---|---|---|---|---|---|---|
| [18] | Mobile networks | SVM, signature and anomaly based methods | DoS attacks | High accuracy to detect normal and anomalous behavior | Only detects DoS attacks | Dataset KDD |
| [19] | Mobile networks | PCA-SVM | Low rate DoS attacks | High detection rate and low FPR and FNR | Only detects DoS attacks | Simulation |
| [20] | In-vehicle networks | One-class SVM | Possible errors in the recordings | The proposed methodology could be applied to several fields | TNR below 77% and precision below 76%. | Dataset from a real vehicle |
| [23] | Mobile networks | MLP for intrusion detection | DoS attacks | High accuracy to detect normal and anomalous behavior | Only detects DoS attacks | Dataset KDD |
| [25] | Heavy duty vehicle system | Gaussian radial basis function neural network | Deception attacks | Can be applied to a variety of nonlinear CPSs | Attacks occur in only one part of the system | Simulation |
| [26] | Solar Farms | Multilayer LSTM network | Integrity attacks | Accuracy, recall, precision and F1 score are above 90% | Attacks occur in only one part of the system | Simulation |
| [37] | Three-tank system | Luenberger Observers (LOs) and Unknown Input Observers (UIOs) | Integrity attacks | Possibility to mitigate the effect of the attack | Attacks occur in only one part of the system. Dependence on threshold selection. | Simulation |
| [38] | Smart grids | Unknown Input Observers (UIOs) | Integrity attacks | Possibility to mitigate the effect of the attack. | Attacks occur in only one part of the system Dependence on threshold selection. | Simulation |
| [39] | Power systems | Unknown Input Observers (UIOs) | Integrity attacks | Possibility to mitigate the effect of the attack | Attacks occur in only one part of the system. Dependence on threshold selection | Simulation |
| [40] | Power systems | Luenberger Observers (LOs) and Unknown Input Observers (UIOs) | Integrity and DoS attacks | Platform for simulating different types of cyber attacks | Detection depends on the selection of the threshold. | Emulation and simulation |
| [41] | Automotive Brake Systems | Recurrente neural networks | Integrity attacks | High accuracy | The attacks are applied on the same part. | Experimental |
| [42] | Industrial Control Systems | 1D CNN and GRU | Integrity attacks | High precision and F1 score. | False alarm rate needs to decrease. | SWaT Dataset |
| [50] | Automated Vehicles | LSTM and CNN | Various | Detecting different single anomaly types. | In some cases the TPR is low. | Experimental. |
| [55] | Heavy-duty gas turbines of combined cycle power plants | Stacked denoising autoencoder | Various | Real time detection, high TPR and low FPR. | Only detects, does not locate. | Simulation and data from real plants. |
| [56] | Automobile Control Network Data | LSTM | Integrity attacks | High TPR and low FPR. | It is required to achieve a practical level to reliably detect anomalies. | Simulation. |
| [59] | Industrial Control System | Genetic algorithms and neural network | Various | High accuracy to locate the sensor under attack. | Metrics, such as F1 score and recall, must be improved. | SWaT Dataset. |
| [60] | Industrial Control System | Deep Neural Networks | Various | Successfully detects the vast majority of attacks with a low level of false positives. | Metrics, such as F1 score and recall, must be improved. | SWaT Dataset. |
| [61] | Industrial Control System | Graphical model-based | Various | High precision. | Metrics, such as F1 score and recall, must be improved. | SWaT Dataset. |
| [62] | Industrial Control System | SVM and Deep Neural Networks | Various | High precision. | Metrics, such as F1 score and recall, must be improved. | SWaT Dataset. |
| [63] | Industrial Control System | LSTM and CNN | Various | High precision. | Metrics, such as F1 score and recall, must be improved. | SWaT Dataset. |
| [64] | Industrial Control System | Lightweight Neural Networks and PCA | Various | Good precision. | Metrics, such as F1 score and recall, must be improved. | SWaT, BATADAL, and WADI Dataset. |
| [65] | Networked Control (Three-tank system) | Resilient Tracking Control | Deception and DoS attacks | A combination of attacks can be taken into account to form a sophisticated and stealthy attack model. | High dependence on knowledge of system parameters. | Simulation and experimental. |
| [66] | Three-tank system | Model-based fault/attack tolerant | Integrity attacks | Determines when the control input is to be updated again, depending on the occurrence of the anomaly. | High dependence on knowledge of system parameters. | Simulation and experimental. |

Based on the review of the related works, it became evident that there are still challenges concerning the detection of cyber attacks within the control systems found in the CPSs. On the one hand, methods must be sought to decrease both the false positive and false negative rates, and to increase the true positive and true negative rates. This will improve the overall performance of these detection systems. It is also evident that the phenomenon of simultaneous attacks has not been addressed in the design of cyber attack detection systems, which is worrying because these situations can occur very often in the real world. Is important to clarify that, within a CPS, there are many points where a cyber-attack can occur and that can cause different consequences in the system. The emphasis of this work seeks to design an architecture that allows detecting and locating attacks that occur between the elements of the physical layer and the controller of a CPS, precisely in attacks that modify or interrupt the sending of data from one element to another. In this way, this paper presents the design of an architecture that explores the potential of convolutional neural networks to extract features and, thus, to determine whether there is an event related to the possibility of a cyber attack occurring. This approach may have a closer approach to the implementation in real cases in which there is a high degree of uncertainty in the process models, since, on many occasions, the way to detect an anomaly or not is done under a process of comparison between estimated values and the real values of the process, which is subsequently evaluated by a threshold. In our proposal, this evaluation is carried out in an intrinsic way by the architecture based on convolutional neural networks, generating a better performance than current works, as well as shows promising results in the detection and isolation of simultaneous attacks.

### 3. Problem Statement

Several control applications supported in these systems can be labeled as safety critical in relation to the fulfillment of strict real time deadlines, associated with the generation of actions from the interaction between the computational systems and the physical systems related to the application, because the non-fulfillment of these requirements can cause irreparable damage to the physical system being controlled, as well as to the people depending on it [70]. Additionally, measurements and control actions can be altered while being transmitted through communication networks, thus requiring new control algorithms or design architectures, which, in the presence of adverse situations, can bring the system to safe and stable states [71,72]. The proposal presented in this work focuses in the detection and isolation of DoS and integrity cyber attacks on CPSs, specifically on the exchange of information between sensors, actuators, and controllers. The approach realized is based in the fault detection and isolation systems for what anomalies are represented as a variation of the system parameters [58]. Then, any control system where its control signals and/or measured variables are susceptible to be attacked can be modeled as a combination of the two models defined in (1) and (2).

$$x(k+1) = Ax(k) + Bu(k) + F_a f_a(k), \tag{1}$$

$$y(k) = Cx(k) + F_s f_s(k), \tag{2}$$

where $x(k)$ represents the state vector, $x(k) \in \mathbb{R}^{n \times 1}$, $y(k)$ is the output vector, $y(k) \in \mathbb{R}^{p \times 1}$, $u(k)$ is the control action, $u(k) \in \mathbb{R}^{m \times 1}$, matrix $A$ is the state matrix, $A \in \mathbb{R}^{n \times n}$, $B$ is the input matrix, $B \in \mathbb{R}^{n \times m}$, $C$ is the output matrix, $C \in \mathbb{R}^{p \times n}$, D is the feedthrough matrix, $D \in \mathbb{R}^{p \times m}$, $F_a = B$, and $f_a = (\Gamma - I)U + U_{f0}$. $\Gamma U$ and $U_{f0}$, represent the effect of a multiplicative anomaly and an additive effect in the control action, respectively. DoS and integrity attacks are visible as anomalies on the control action. If the $i$-th control action is attacked, then the matrix $F_a$ corresponds to the $i$-th column of the matrix $B$, and $f_a$ corresponds to the magnitude of the attack that directly affects the controller.

Similarly, if the *i*-th sensor is attacked, the matrix $F_s$ is the *i*-th row of the matrix $C$, and the vector of attacks is $f_s$, which represents the magnitude of the effect produced in the *i*-th sensor.

The problem with traditional methods based on mathematical models that describe the behavior of the system is that these models are dispensable of the complete knowledge of the system parameters, and the adaptation in real conditions can cause the overall performance to decrease. Because of this, we intend to address this problem from models based on artificial neural networks, precisely in one-dimensional convolutional neural networks, which have shown very promising results in fields where patterns are sought to identify a class.

*Modeling of the Cyber Attack*

Measurements of process signals and control action values are critical to the proper functioning of a control system, and its modification by cyber attacks can produce instability in the control system [73,74]. A cyber attack by data manipulation is called an integrity attack, modeled by (3), and an attack that results in a prolonged loss of these signals is called a type DoS attack, which is modeled by (4).

$$\overline{y}_i(k) = y_i(k) + y_i(k)^a,\tag{3}$$

$$\overline{y}_i(k) = y_i(k)_{t_{s-1}},\tag{4}$$

where $\overline{y}_i(k)$ corresponds to the sensor measurement that reaches the controller in the k-time, $y_i(k)$ corresponds to the sensor measurement before being transmitted to the controller in the k-time, and $y_i(k)^a$ is a vector injected by the attackers which changes the $y_i(k)$ measure in the k-time. $y_i(k)_{t_{s-1}}$ corresponds to the measurement before the start of the DoS attack. The time interval for the occurrence of the attack is defined by $\tau_a = [t_s\ t_e]$.

For the development of the proposal, it is assumed that any sensor can be affected by any type of attack, integrity, or DoS. Additionally, the attacks may occur at any time in various parts of the system. The last premise is significant to note because simultaneous attacks are less discussed in previous works; thus, depending on the type of attack carried out on the system, output (2) may take the form of (3) and/or (4).

## 4. Attack Detection and Isolation Method

In the context of this work, most cyber attack detection methods use the available data to develop a model that determines the usual behavior of the system. Then, by a comparison between the estimated outputs of the model and the actual process outputs, determination of if the behavior of the system is normal or if a cyber attack is taking place. To isolate the attack, which is nothing more than locating the part of the system that is being affected directly by the cyber attack, decoupled models of the system are developed that are susceptible only to cyber attacks that occur in specific parts of the system.

The procedure to perform this task can be grouped into three steps. Firstly, the generation of a residual signal is realized, and this process consists of comparing the measured output with an estimated output. This signal is denoted as residual signal, $res(k)$, this is described in (5).

$$res(k) = y(k) - \hat{y}(k),\tag{5}$$

where $y(k)$ are the set of output measures of the actual process, and $\hat{y}(k)$ are the set of outputs estimated. The second step corresponds to the evaluation of the residual; in this case, a comparison of the residuals is made with a predefined threshold, as is shown in (6).

$$|res(k)| > \tau_{thresholds}.\tag{6}$$

The thresholds are obtained from data in which the attacks have been presented, thus allowing their detection and isolation. Finally, a decision-making process is carried out through indicators.

These steps involve the use of residuals that should take values close to 0 in situations where the system is not being attacked. On the other hand, when an attack is present, the residual signals must have values other than 0.

Although a single residual signal can alert or detect a cyber attack, a set of residuals is required to isolate it. Then, to locate the origin of the cyber attack, it is necessary that some residues be sensitive only for a particular part of the system. The above implies that the set of residuals must be independent of other cyber attacks defined. In this way, to isolate a cyber attack, a structured set of residuals is considered, where each residual vector can be used to detect a cyber attack in a specific place of the system.

In the architecture model proposed in this work, it is emphasized that second step will be an implicit step because the architecture based on artificial neural networks will interpret the input data generating intrinsic characteristics that will allow the evaluation to detect and isolate the attacks.

*Architecture Proposed for the Detection and Isolation of Cyber Attacks in CPS*

The architecture proposed is presented in Figure 2. This architecture includes a prediction model which uses an input dataset $x_0, x_1, \ldots, x_{k-1}$ to estimate outputs $\hat{y}_1, \hat{y}_2, \ldots, \hat{y}_k$ (these datasets will depend specifically on the type of data available from the process), and these values are used to obtain the residual signal $res(k)$, as is shown in (5). These signals are used by a classifier to detect anomalies presents in the process.

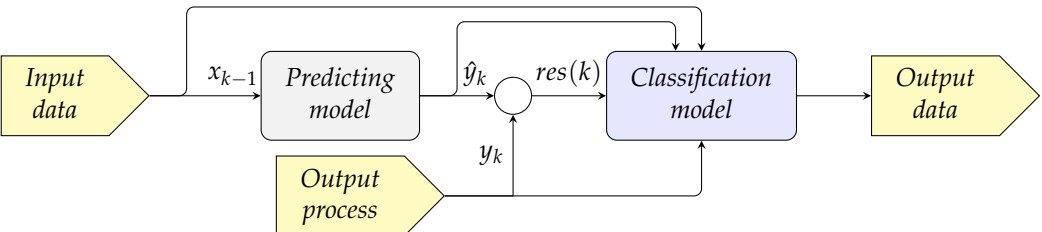

**Figure 2.** General architecture model to detect and isolate cyber attack.

As the characteristics of the signals in a specific process are different then values with different magnitude could affect the classifier training procedure, therefore, all input data to the classifier are normalized using its mean and standard deviation to obtain a z-score for each one as is shown in (7).

$$z = |x - \mu| / \sigma, \tag{7}$$

where $x$ are the input data, $\mu$ is the mean, and $\sigma$ is the standard deviation.

Although the architecture presented is general, it is a base for selecting different types of machine learning for the prediction and classification stages. The idea is to use deep neural networks to extract patterns that allow the detection of cyber attacks (such as LSTM or CNN 1-dimensional). As was not included a method to find spatial-temporal correlations to detect cyber attacks, it is expected that neural networks will be able to carry out this task implicitly.

The architecture can be detailed as follows for a specific CPS, shown in Figure 3. A model of the dynamics of the process generates the outputs signals $x(k)_s$ which correspond to the reconstruction of all the states (it is assumed that the outputs are the process states or some linear combination of them, although it can be extended to non-linear cases). In order to isolate the attack, there is a set of neural network models that relate the process states with their respective control actions for generate states that are decoupled from each other $(x(k)_{d_{1,2,\ldots x}})$; in this way, it is possible to isolate the attack in a way equivalent to the use of UIOs, but with the advantage that neural networks allow addressing the uncertainty in the representations. With this set of neural networks, $res(k)$ is generated.

Detection and isolation functions are implemented using artificial neural networks, which use the process states $x(k)$, the control actions $u(k)$, the reference signals $r(k)$, the residual signals $res(k)$, and the signals generated by the predicting model.

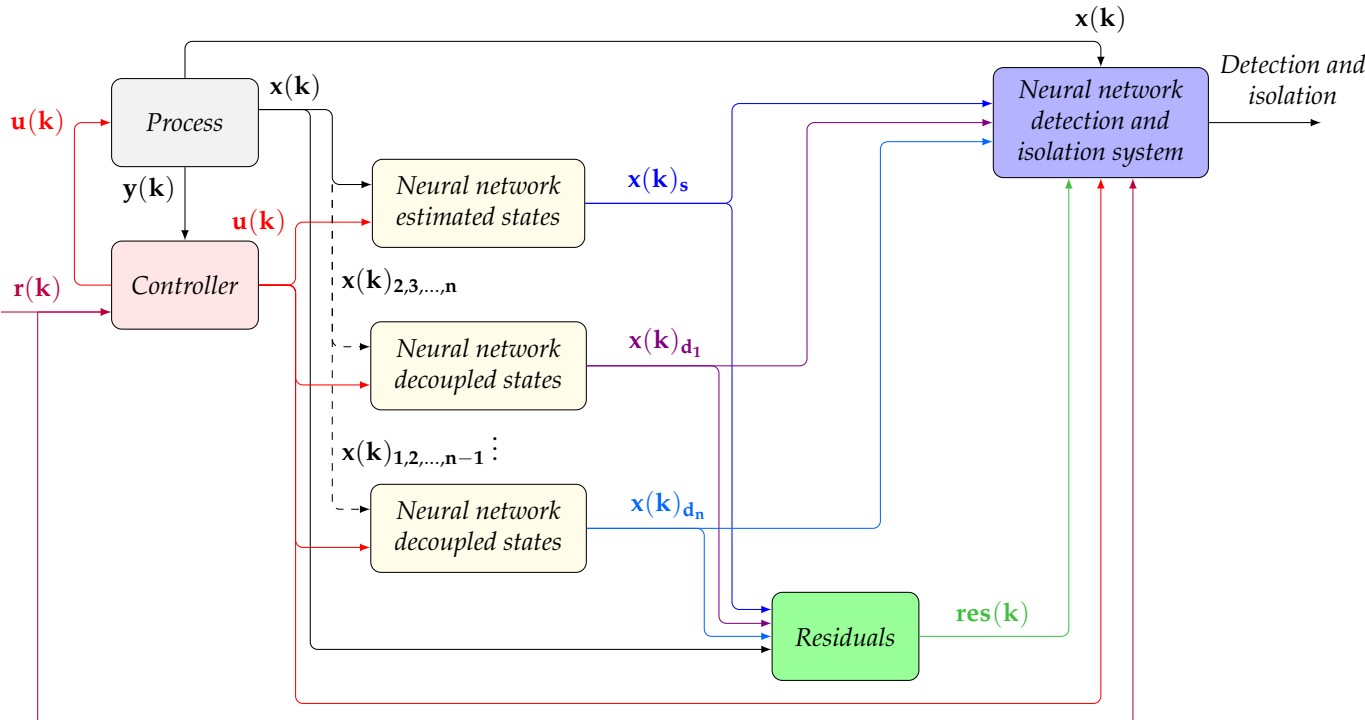

**Figure 3.** Architecture based on neural networks for the detection and isolation of the cyber attack.

Mean squared error (MSE) [75] is adopted as the model's loss function to train the predicting model.

$$MSE = \frac{1}{n} \sum_{i=1}^{n} (x_i - \hat{x}_i)^2, \tag{8}$$

where $n$ is the amount of data, $x_i$ is the real state, and $\hat{x}_i$ is the estimated state. For the classifier, the cost function categorical crossentropy (CCE) is used [76] because it is a single-label multi-class classification problem.

$$J_{CCE} = - \sum_{q=1}^{l} \sum_{k=1}^{p} d_{qk} \log(y_{qk}). \tag{9}$$

With $p$ classes, training data size of $l$, the input of $x_q$, where $q = 1, 2, \ldots, l$ and $y_{qk}$ $(0 \leq y_{qk} \leq 1), k = 1, 2, \ldots, p$ is the estimated probability that belongs to class $k$, and $d_{qk}$ (0 or 1) becomes the given label (9).

## 5. Case Studies and Results Analysis

Two test benches were used to evaluate the performance of the proposed architecture, the SWaT dataset [77,78] and an interconnected tank [58].

### 5.1. Secure Water Treatment Dataset-SWaT

This dataset was completed by the Singapore University of Technology and Design to provide researchers with data collected from a complex and realistic ICS environment. The testbed is a fully operational scale water treatment plant that produces purified water. SWaT is composed of six main processes corresponding to the physical and control components of the water treatment plant; each stage (from P1 to P6) is equipped with a certain number of sensors and actuators. The sensors include flow meters, water level meters, conductivity, and pH analyzers, among others, while the actuators consist of pumps that transfer water from one stage to another, pumps that dose chemicals, and valves that control inlet flow. The process is not circular, and P6 water is removed. Sensors and actuators in each stage

are connected to the corresponding PLC (programmable logic controller). This process is shown in Figure 4.

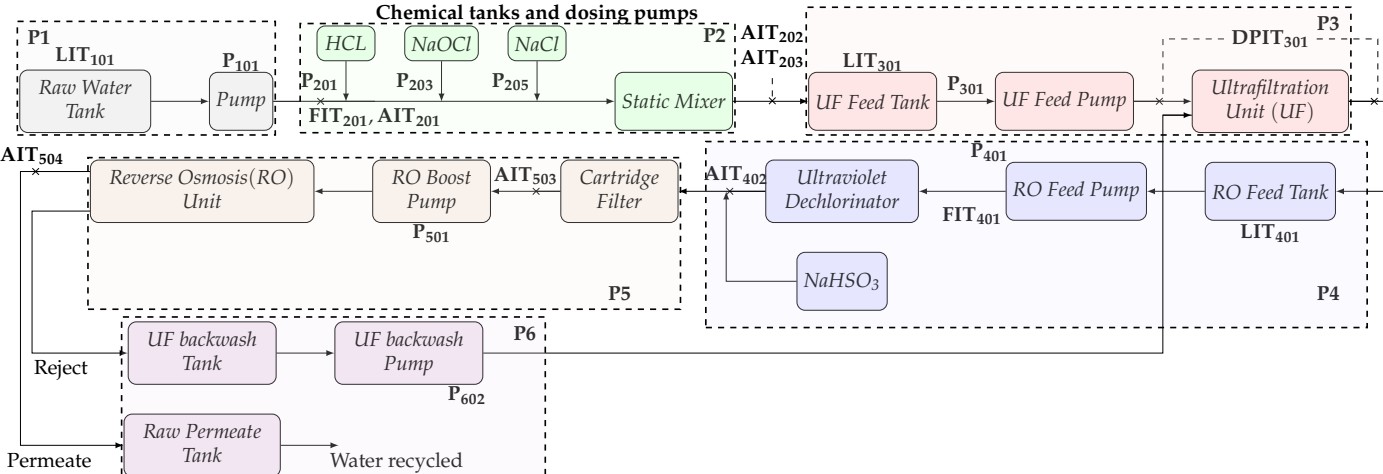

**Figure 4.** SWaT testbed processes overview [57].

Stage P1 controls the flow of raw water by opening or closing a motorized valve that is connected to the inlet of tank T101. By means of the P101 pump, water starts flowing from T101 through the chemical dosing station in stage P2 and is followed by the ultrafiltration (UF) process located in stage P3, which eliminates unwanted materials. Similarly, the feed pump in stage P3 is responsible for supplying the water to the ultrafiltration unit. In the P5 stage, inorganic impurities are separated by a reverse osmosis process. The output of the reverse osmosis process is stocked in the permeate tank of stage P6 for its distribution. The P6 stage is also controlling the cleaning of the ultrafiltration membranes in P3 by the backwashing process. Every certain period of time, the backwash process is triggered by turning on the backwash pump and is accomplished in under one minute. The backwash process can alternatively be started by a PLC when the differential pressure sensor value increases above 0.4, which means that the UF membranes are choked [57,78].

### 5.1.1. Dataset Description

Training Dataset 1 and Training Dataset 2 were used. The first one corresponds to data collected under normal operating conditions. This dataset was released on November 20, 2016 and was generated from a one-year long simulation. The second dataset corresponds to situations when attack scenarios are generated. This dataset with partially labeled data was released on 28 November 2016. The dataset is around six months long and contains several attacks, as shown in Table 2.

**Table 2.** Attacks featured in Training dataset 2 [78].

| ID | Duration (Hours) | Attack Description | SCADA Concealment |
|----|------------------|--------------------|-------------------|
| 1 | 50 | Attackers change L_T7 thresholds (which controls PU10/PU11) by altering SCADA transmision to PLC9. Low levels in T7. | Replay attack on L_T7. |
| 2 | 24 | Like Attack # 1. | Like Attack # 1 but replay attack extended to PU10/PU11 flow and status. |
| 3 | 60 | Attackers alter L_T1 readings sent by PLC2 to PLC1, which reads a constant low level and keeps pumps PU1/PU2 ON. Overflow in T1 | Polyline to offset L_T1 increase. |

**Table 2.** *Cont.*

| ID | Duration (Hours) | Attack Description | SCADA Concealment |
|----|------------------|--------------------|--------------------|
| 4 | 94 | Like Attack # 3. | Replay attack on L_T1, PU1/PU2 flow and status, as well as pressure at pumps outlet. |
| 5 | 60 | Working speed of PU7 reduced to 0.9 of nominal speed causes lower later levels in T4. | |
| 6 | 94 | Like Attack # 5, but speed reduced to 0.7. | L_T4 drop concealed with replay attack. |
| 7 | 110 | Like Attack # 6. | Replay attack on L_T1, as well as PU1/PU2 flow and status. |

### 5.1.2. Data Preparation and Model Training

The data from the first dataset is used to generate a model corresponding to the "Predicting model" block shown in Figure 2. The architecture proposed in this case is based on a 1D CNN model, as shown in Figure 5.

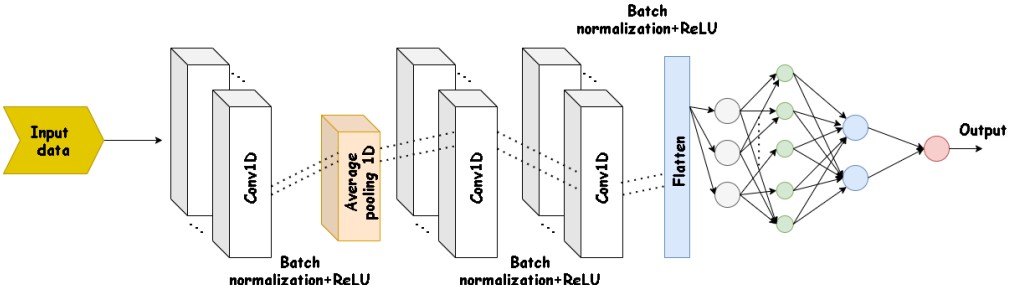

**Figure 5.** Prediction model for SWaT dataset.

The input data is composed of 43 characteristics compounded mainly by sensors measurements, states of the pumps, and valves positions. The first convolution layer consists of 2 filters, and the kernel size is 3. The 1D average pooling layer has a stride of 2 and the same padding; the second convolution layer has 20 filters and a kernel size of 20; the last convolution layer is composed of 10 filters and a kernel size of 5. Finally, a fully connected layer is used with a 43 neurons layer and a neuron in the output layer, all with linear activation functions. Additionally, the batch normalization layer is added with ReLU activation in various parts of the network. The loss function used was MSE, and the optimizer was the stochastic gradient descent with momentum. For training, a maximum of 40 epochs was available with an initial learning rate of 0.001. In this case, 30% of the data was used to validate, and 70% of the data to train.

The parameters of the layers for this network were found in such a way that the lowest possible MSE will be achieved. Increasing the number of layers, neurons, filter size, or number of filters did not correspond to a significant improvement performance.

The second dataset was used for the classification process; it is composed of 4177 data, of which 3685 data correspond to normal operating conditions, 50 belong to the first attack scenario, 24 correspond to the second attack scenario, 60 to third and fifth attack, 94 to fourth and sixth attack, and 110 to the seventh scenario. As can be seen in Figure 6a, this dataset is unbalanced and would then generate problems to the classifier. The bar centered at 0 corresponds to normal operating conditions, while the other corresponds to the different attack scenarios which are shown in the ID column of Table 2. It could affect the algorithms in relation to the minority classes. To address this situation, initially, methods, such as Random Oversampling and Undersampling, were used for imbalanced classification without obtaining satisfactory results. For this reason, the approach shown in Reference [79] was followed. This proposal is a modification of temporal data determined by optimal sequences that are aligned with the original data, thus generating new time-

synthesized data to the training dataset. The distribution of the different classes for the new dataset to be used is shown in Figure 6b. Although it is observed that it is an unbalanced dataset, the amount of data generated from the attack scenarios was increased, and the performance was improved.

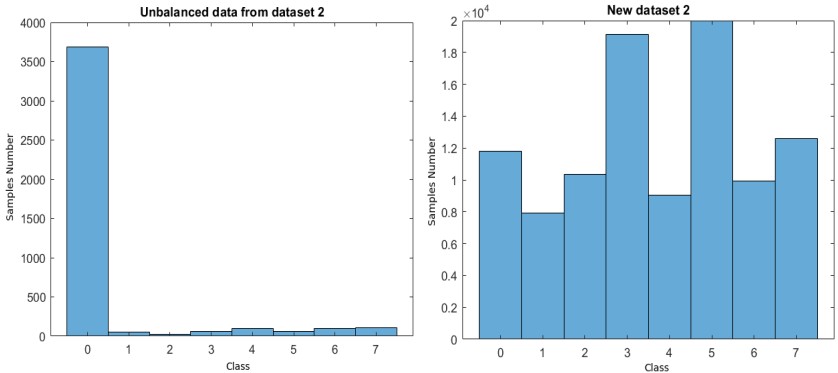

(**a**) Original SWAT dataset distribution. (**b**) New SWAT dataset distribution.

**Figure 6.** SWAT dataset distribution.

This new dataset was used to estimate the outputs using the architecture shown in Figure 5, which were compared with the usual process variables to obtain the residual signal.

The input data for the classifier whose architecture is shown in Figure 7 are: the estimated outputs, the process variables, and the corresponding residuals. This corresponds to the "Classification model" block shown in Figure 2 and was implemented by a group of cascaded convolutional layers with a batch normalization layer with ReLU activation function between them. The number of convolutional layers selected was five, obtaining a higher accuracy than 90%. The number of filters implemented from the input to the fully connected layer were 128, 64, 32, 16, and 8, respectively. The kernel size in each one was 10. The fully connected layer is composed of eight neurons in its input layer with linear activation function, while the last layer has eight neurons with softmax activation functions corresponding to the 7 attacks and the usual operation scenarios.

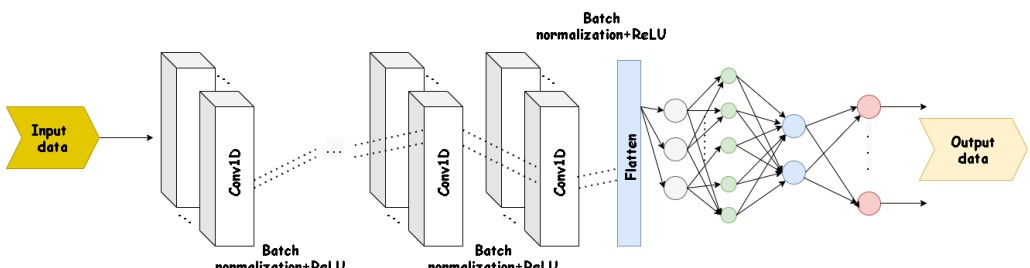

**Figure 7.** Classification model for SWaT dataset.

The loss function used was CCE, and the optimizer used was stochastic gradient descent with momentum. For the training, a maximum of 4 epochs was available, with an initial learning rate of 0.0001. For training, a maximum of 4 epochs was available, with an initial learning rate of 0.0001, and 30% of the dataset was used to validate, while 70% was used to train.

### 5.1.3. Evaluation Metrics

The metrics considered in this work were true positives (*TP*), false positives (*FP*), true negatives (*TN*), and false negatives (*FN*). In order to evaluate the performance of the architecture proposed, the following metrics were used: precision, accuracy, recall

(sensitivity or TPR), F1 score, and true negative rate or specificity (*TNR*). These metrics were calculated as follows:

$$Precision = \frac{TP}{TP + FP}, \tag{10}$$

$$Accuracy = \frac{TP + TN}{TP + TN + FP + FN}, \tag{11}$$

$$Recall = \frac{TP}{TP + FN}, \tag{12}$$

$$F1\ Score = \frac{2TP}{2TP + FP + FN} = 2\frac{Precision \times Recall}{Precision + Recall}, \tag{13}$$

$$TNR = \frac{TN}{FP + TN}. \tag{14}$$

Additionally, the ROC (Receiver Operating Characteristics) and Precision-Recall Curves were considered.

### 5.1.4. Analysis of Results of SWaT Case

The results obtained for this dataset are shown in this section. The training and recovering results are carried out in MATLAB software. Figure 8 shows the confusion matrix for each of the available classes. From these results, the metrics defined in the previous section are obtained and are presented in Table 3.

**Figure 8.** Confusion matrix for SWaT dataset.

**Table 3.** Summary of metrics.

|         | Accuracy | Precision | Recall | F1 Score | TNR  |
|---------|----------|-----------|--------|----------|------|
| Class 0 | 0.97     | 0.81      | 0.97   | 0.88     | 0.98 |
| Class 1 | 0.99     | 0.99      | 0.99   | 0.99     | 0.99 |
| Class 2 | 0.99     | 0.98      | 0.91   | 0.94     | 0.99 |
| Class 3 | 0.99     | 0.99      | 0.95   | 0.97     | 0.98 |
| Class 4 | 0.99     | 0.94      | 0.94   | 0.94     | 0.99 |
| Class 5 | 0.99     | 0.98      | 0.97   | 0.97     | 0.98 |
| Class 6 | 0.99     | 0.99      | 0.99   | 0.99     | 0.99 |
| Class 7 | 0.98     | 0.95      | 0.89   | 0.92     | 0.98 |

Class 0 corresponds to the usual operation, while class 1 to 7 are the different attacks scenarios shown in Table 2. It is observed that accuracy is high in all cases. The above shows a high percentage ratio of samples correctly classified by our model. On the other hand, for precision, it is observed that all attack scenarios present a score above 0.94, which means that a lot of data was correctly classified in the different attack scenarios. Similarly, the recall scores are above 0.91 in the majority of classes, which allows minimizing the false alarm rate. Finally, the F1 score shows scores above 0.92. The high rate of TNR in each of the classes is highlighted, which means that FPR is low.

The ROC and Precision-Recall Curves shown in Figure 9a,b present an appropriate performance, indicating that the model has a good capability to distinguish different classes.

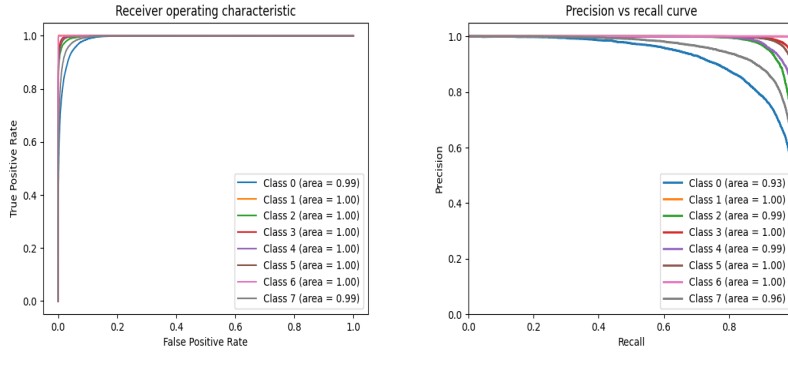

(**a**) ROC curve.  (**b**) Precision-Recall Curve.

**Figure 9.** ROC and Precision-Recall Curves for SWaT dataset.

Table 4 presents a comparison of the proposal presented in this is paper with other methods. In the recall and F1 score metrics, the proposed method presents a better performance related to the other methods. For values of precision and accuracy, the proposed method is above in almost all cases, except for the last two methods, which exceed it by a score margin of 0.04. However, the performance of the F1 score metric is high, indicating that a satisfactory and reliable class detection was obtained.

**Table 4.** Summary of the results and performance comparison on the SWaT dataset.

| Method | Accuracy | Precision | Recall | F1 Score |
|---|---|---|---|---|
| Proposed | 0.95 | 0.95 | 0.95 | 0.95 |
| SVM [59] | - | 0.93 | 0.70 | 0.79 |
| RNN [59] | - | 0.94 | 0.70 | 0.80 |
| 1D CNN [60] | - | 0.96 | 0.80 | 0.87 |
| TABOR [61] | 0.95 | 0.86 | 0.79 | 0.82 |
| STAE-AD [63] | - | 0.96 | 0.82 | 0.88 |
| AE [64] | - | 0.89 | 0.80 | 0.84 |
| AE Frequency [64] | - | 0.92 | 0.83 | 0.87 |
| LSTM [62] | - | 0.98 | 0.68 | 0.88 |
| One Class SVM [62] | - | 0.93 | 0.70 | 0.80 |
| SDA+1D CNN+ LSTM [42] | 0.99 | 0.99 | 0.85 | 0.91 |
| SDA+1D CNN+ GRU [42] | 0.99 | 0.99 | 0.85 | 0.92 |

*5.2. Interconnected Tank Testbed*

This testbed has been used extensively to test proposals to detect anomalies [37,65–69]. The hydraulic system consists of three identical cylindrical tanks with equal cross-sectional area $S$, as shown in Figure 10. These tanks are connected by two cylindrical pipes of the same cross-sectional area, denoted $S_n$, and have the same outflow coefficient, denoted $\mu_{13}$ and $\mu_{32}$. The nominal outflow located at tank 2 has the same cross-sectional area as the coupling pipe between the cylinders, but a different outflow coefficient, denoted $\mu_{20}$. The

inlet flow of the tanks comes from two pumps, with a flow rate, $q_1$ and $q_2$. A digital/analog converter is used to control each pump. A piezo-resistive differential pressure sensor carries out the necessary level measurement. The idea of the system is to maintain the height levels of the fluid stored in tanks 1 and 2 at a particular operating point.

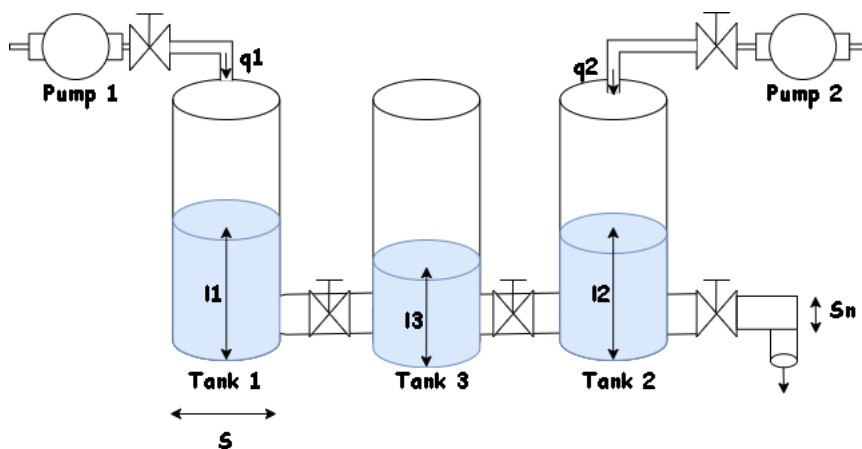

**Figure 10.** Schematic diagram of the three-tank system.

The parameters are shown in Table 5, and the mathematical model is presented in (15)–(17) [58].

$$\frac{dl_1(t)}{dt} = (q_1(t) - q_{13}(t))/S$$
$$\frac{dl_2(t)}{dt} = (q_2(t) + q_{32}(t) - q_{20}(t))/S, \tag{15}$$
$$\frac{dl_3(t)}{dt} = (q_{13}(t) - q_{32}(t))/S$$

$$q_{mn}(t) = \mu_{mn}S_p sign(l_m(t) - l_n(t))\sqrt{2g|l_m(t) - l_n(t)|} \ (m, n = 1, 2, 3 \ \forall \ m \neq n), \tag{16}$$

$$q_{20}(t) = \mu_{20}S_p\sqrt{2gl_2(t)}. \tag{17}$$

**Table 5.** Parameters value of the three-tank system.

| Variable | Symbol | Value |
| --- | --- | --- |
| Tank cross sectional area | $S$ | $0.0154 \text{ m}^2$ |
| Inter tank cross sectional area | $S_n$ | $5 \times 10^{-5} \text{ m}^2$ |
| Outflow coefficient | $\mu_{13} = \mu_{32}$ | 0.05 |
| Outflow coefficient | $\mu_{20}$ | 0.675 |
| Maximum flow rate | $q_{imax}(i \in [1\,2])$ | $1.2 \times 10^{-4} \text{ m}^3/s$ |
| Maximum level | $l_{jmax}(j \in [1\,2\,3])$ | 0.62 |

5.2.1. Dataset Generation

Assuming that $l_1 > l_2 > l_3$, a linear approximation was established around an equilibrium point $(U_0, Y_0)$ using Taylor series expansion. The linearized system is described by a discrete state space representation with a sampling period of $T_s = 1s$. This is shown in (18).

$$\begin{aligned} x(k+1) &= Ax(k) + Bu(k) \\ y(k) &= Cx(k) \end{aligned}. \tag{18}$$

The states $x(k)$ correspond to the fluid level of the tanks.

The purpose of this study is to control system around the operating point $(U_0, Y_0)$, as is shown in (19).

$$Y_0 = [0.4\ 0.2\ 0.3]^T\ (m)$$
$$U_0 = [0.35 \times 10^{-4}\ 0.375 \times 10^{-4}]^T\ (m^3/s) \tag{19}$$

A tracking control problem was considered in this study case, where the desired outputs $y = [l_1\ l_2]^T$ are required to track references. The state feedback pole assignment technique was used. Thus, a feedback gain matrix $K$ was designed, so that the closed-loop eigenvalues of the augmented system are equal to $[0.92\ 0.97\ 0.9\ 0.95\ 0.94]$. MATLAB software was used to find the matrices $A$ and $B$, as well as the controller gains. The values can be observed in (20)–(22).

$$A = \begin{bmatrix} 0.9888 & 0.0001 & 0.0112 \\ 0.0001 & 0.9781 & 0.0111 \\ 0.0112 & 0.0111 & 0.9776 \end{bmatrix}, \tag{20}$$

$$B = \begin{bmatrix} 64.5687 & 0.0014 \\ 0.0014 & 64.2202 \\ 0.3650 & 0.3637 \end{bmatrix}, \tag{21}$$

$$K = [K_1\ |K_2] = 10^{-4} \left[ \begin{pmatrix} 21.6 & 3 & -5 \\ 2.9 & 19 & -4 \end{pmatrix} \middle| \begin{pmatrix} -0.95 & -0.32 \\ -0.3 & -0.91 \end{pmatrix} \cdot \right] \tag{22}$$

In order to construct the dataset for detecting attacks, the scheme shown in Figure 11 was implemented, which has modules to obtain measurements of the process variables, as well as the control actions applied by the actuators. An Ethernet was used as a control network. This representation is equivalent to boxes "Process" and "Controller" in the architectures presented in Figure 3.

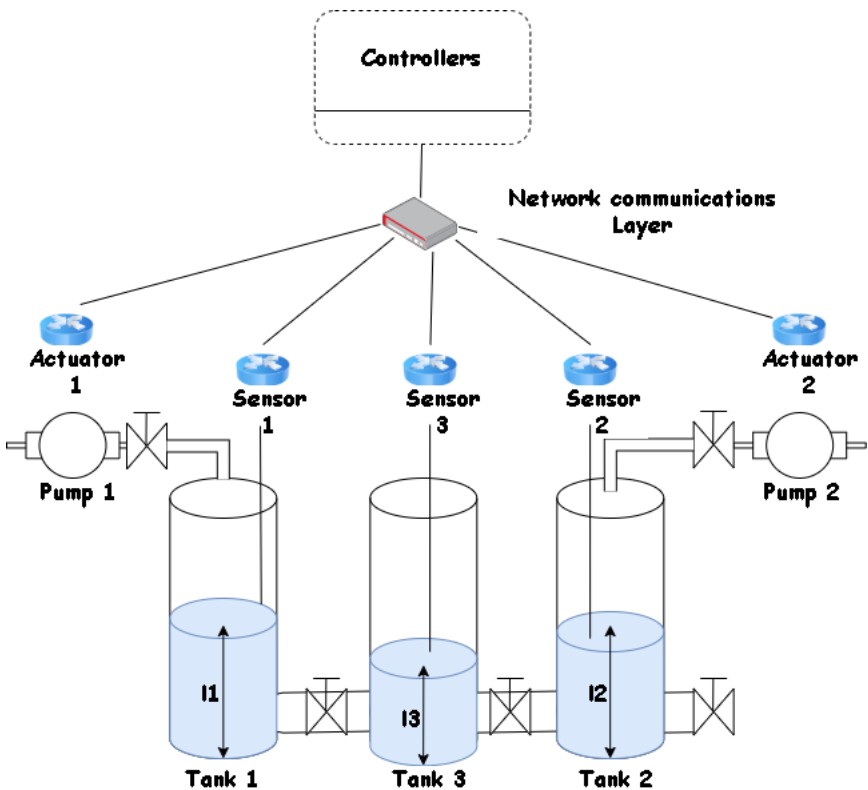

**Figure 11.** Interconnected tank testbed.

Two datasets were generated. The first one is a dataset in normal operations to determine a model that estimates the system outputs. The second one includes cyber attacks on sensors 1 and 2. These cyber attacks can be integrity or DoS attacks.

In both cases, 499,000 samples were generated. The system references range between 0.35 m and 0.45 m for $l_1$, and between 0.185 m and 0.25 m for $l_2$. The time intervals were defined randomly with a uniform distribution and reference changes every 500 s to 850 s.

The cases are shown in Table 6. Case 0 corresponds to operation without attacks. The following cases correspond to situations in which integrity or DoS cyber attacks can be generated on any sensor, following the models described by the Equations (3) and (4). In cases 1 to 4, only one cyber attack is generated every time, while cases 5 to 8 correspond to simultaneous attacks.

**Table 6.** Cases raised.

| Case | Description |
| --- | --- |
| Case 0 | Normal operation |
| Case 1 | Integrity attack on sensor 1 |
| Case 2 | Integrity attack on sensor 2 |
| Case 3 | DoS attack on sensor 1 |
| Case 4 | DoS attack on sensor 2 |
| Case 5 | Integrity attack on sensor 1 and DoS on sensor 2 |
| Case 6 | Integrity attack on sensor 2 and DoS on sensor 1 |
| Case 7 | Integrity attack on sensor 1 and 2 |
| Case 8 | DoS attack on sensor 1 and 2 |

The time intervals in which cyber attacks occur were defined such that the dataset was balanced, so it were defined randomly and uniformly distributed. The integrity attacks were implemented by changing the modified variable in a range of 5% to 8% of its measured value. This range of values depends on the sensitivity of the system since there will be particular processes where the effect of the variation of the measurements in a given range does not has as much impact as in others. All cases presented correspond to the classes that the classifier will identify. The distribution of these data is shown in Figure 12.

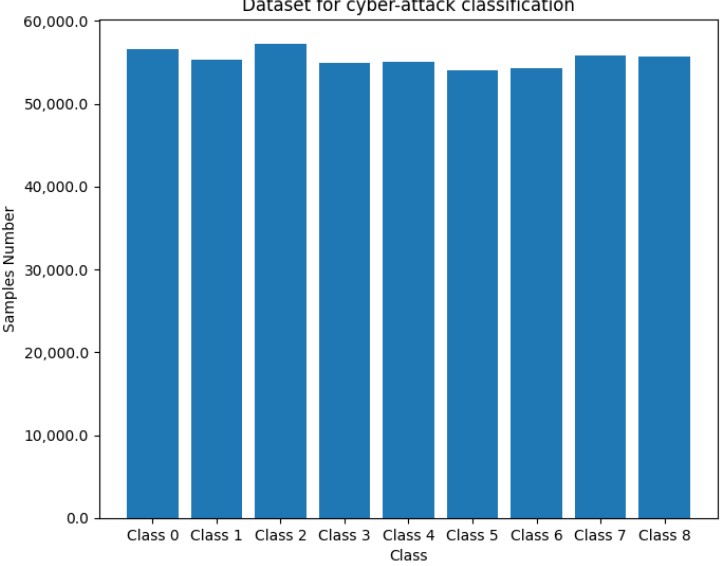

**Figure 12.** Dataset for cyber attack classification.

### 5.2.2. Model Training

Figure 3 presents the architecture implemented. The first model generates the process states estimate, while two more models were obtained to reconstruct independent states $x_1$ and $x_2$, according to those states susceptible to cyber attack.

The first network has the architecture shown in Figure 13. Its input data is composed of five characteristics, which are composed of the measurements of the sensors and the control actions corresponding to vector (23):

$$\begin{aligned}
input\ data =&[x_1(1),\ldots,x_1(k-1),x_2(1),\ldots,x_2(k-1),x_3(1),\ldots,x_3(k-1),\\
&u_1(1),\ldots,u_1(k-1),u_2(1),\ldots,u_2(k-1)]^T
\end{aligned}. \tag{23}$$

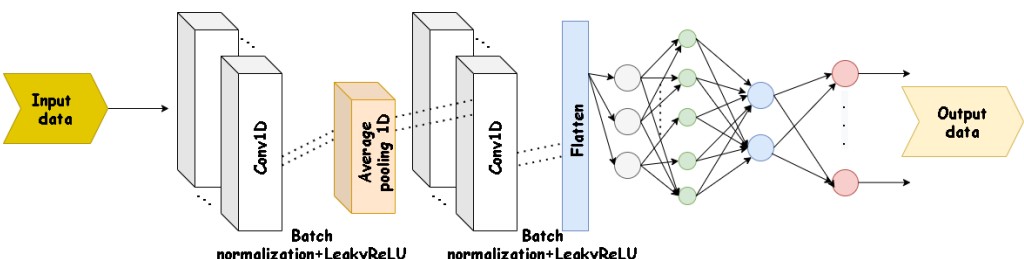

**Figure 13.** Model to estimate all states.

The model has three outputs corresponding to the states of the process. The vector to be reconstructed is (24):

$$\begin{aligned}
output\ data\ 1 &= \hat{x}_1 = [x_1(2),\ldots,x_1(k)]^T\\
output\ data\ 2 &= \hat{x}_2 = [x_2(2),\ldots,x_2(k)]^T,\\
output\ data\ 3 &= \hat{x}_3 = [x_3(2),\ldots,x_3(k)]^T
\end{aligned} \tag{24}$$

where $k$ is the number of samples. This model has two convolutional layers, one average pooling 1D layer between the convolutional layers, and one fully connected layer. The first convolutional layer has a kernel size of 5 and has eight filters, while the second layer has a kernel size of 3 with 16 filters. Each of these layers has hyperbolic tangent activation function. Between previous layers, there is an average pooling 1D layer with a pool size of 2 and strides of 2 with same padding. Between the convolutional layers and the fully connected layer, there is a batch normalization layer with Leaky ReLU type activation function. In the fully connected layer, there is an input layer of 48 neurons and an output layer composed of 3 neurons with a linear activation function to estimate the corresponding states. The loss function used was MSE, and the optimizer used was Adam. For training, a maximum of 4 epochs and a batch size of 10 were available with initial learning rate of 0.01. To train the model, 30% of the data was used to validate, and 70% to train. The various parameters of the layers of this network were found in such a way that the lowest possible MSE will be achieved, it was 0.000067. Increasing the number of layers, neurons, filter size, or number of filters did not correspond to a significant improvement to the proposed architecture.

The second and third networks have the architecture shown in the Figure 14.

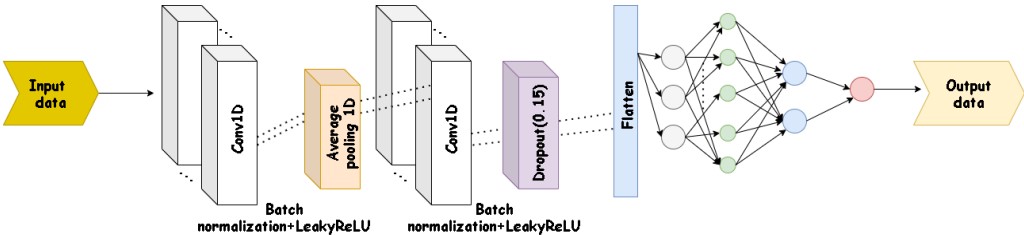

**Figure 14.** Model to estimate the decoupled states.

The input data for the second architecture is composed of four characteristics corresponding to the measurements of the sensors and the control actions, as is presented in vector (25):

$$input\ data = [x_2(1), \ldots, x_2(k-1), x_3(1), \ldots, x_3(k-1),$$
$$u_1(1), \ldots, u_1(k-1), u_2(1), \ldots, u_2(k-1)]^T. \quad (25)$$

The model generates an estimated uncoupled output for the first state as (26):

$$output\ data = \hat{x}_{1d} = [x_1(2), \ldots, x_1(k)]^T, \quad (26)$$

where *k* is the number of samples. This model has two convolutional layers and one fully connected layer. The first convolutional layer has a kernel size of 4 and has 8 filters, while the second layer has a kernel size of 2 with 16 filters. Each of these layers has the hyperbolic tangent activation function. Between these layers, there is an average pooling 1D layer with a pool size of 2 and a stride of 2 with the same padding. Between the convolutional layers and the fully connected layer, there is a batch normalization layer and a Leaky ReLU type activation function. Before the fully connected layer, a dropout layer (0.15) was added. In the fully connected layer, there is an input layer of 32 neurons and an output layer composed of 1 neuron with linear activation function to estimate the corresponding state. The loss functions used was MSE, and the optimizer used was Adam. For training, a maximum of 4 epochs and a batch size of 10 was available with initial learn rate of 0.01. Seventy percent of the data was used to train the model, and 30% to validate it. The various parameters of the layers of this network were found in such a way that the lowest possible MSE will be achieved, and it was 0.00047. Increasing the number of layers, neurons, filter size, or number of filters did not correspond to a significant improvement to the proposed architecture.

Finally, the structure used to estimate the second uncoupled state of the process is shown in (27) and (28). The respective MSE for this case was 0.000031.

$$input\ data = [x_1(1), \ldots, x_1(k-1), x_3(1), \ldots, x_3(k-1),$$
$$u_1(1), \ldots, u_1(k-1), u_2(1), \ldots, u_2(k-1)]^{T'} \quad (27)$$

$$output\ data = \hat{x}_{2d} = [x_2(2), \ldots, x_2(k)]^T \quad (28)$$

The architecture proposed for the classifier of the cyber attack is similar to that shown in Figure 7. It is composed of three convolutional layers whose activation function is hyperbolic tangent. The first convolutional layer has a kernel size of 15 with several 80 filters. The second and third convolutional layers have the same kernel size, but the number of filters is 60 and 30, respectively. There is also a batch normalization layer with Leaky ReLU activation function. Finally, a fully connected layer is used with an input layer of 25 neurons and an output layer with nine neurons corresponding to the established classes above. The last layer uses the softmax function. The loss function used was CCE, and the optimizer used was stochastic gradient descent with momentum. For training, a maximum of 1000 epochs was established, with a batch size of 10 and initial learning rate of 0.0001. For model training, 30% of the data was used to validate, and 70% to train. The input data is (29):

$$input\ data = [x_1, x_2, x_3, \hat{x}_1, \hat{x}_2, \hat{x}_3, \hat{x}_{1d}, \hat{x}_{2d}, q_1, q_2, res, res_1, res_2]^T, \quad (29)$$

where $x_1, x_2, x_3$ correspond to the real variables of the process; $\hat{x}_1, \hat{x}_2, \hat{x}_3$ are the outputs estimated by the architecture shown in Figure 13; $\hat{x}_{1d}$ and $\hat{x}_{2d}$ correspond to the decoupled states estimated by the architecture of the Figure 14, $q_1$ and $q_2$ are the process references, and $res$, $res_1$, and $res_2$ are the residual signals obtained by comparing the real process states with the estimated states, and the individual comparison between the first two real process states and the estimated decoupled states, respectively.

Figure 15a,b present the evolution of the cost function and the accuracy metric obtained during the classifier training procedure.

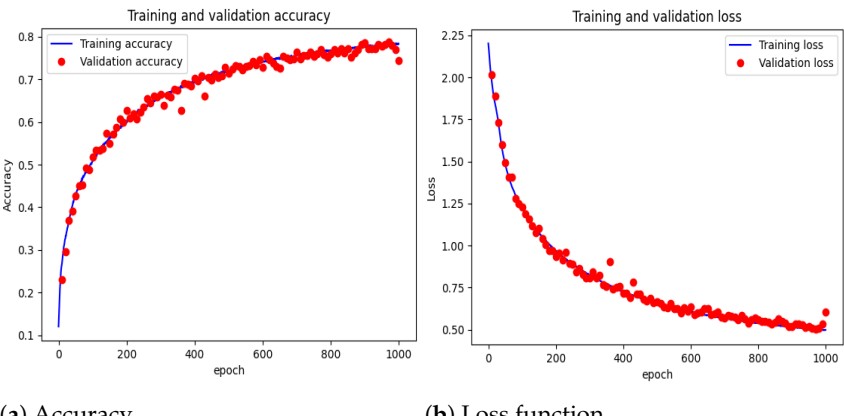

(**a**) Accuracy.                                    (**b**) Loss function.

**Figure 15.** Training and Validation: accuracy and loss function through epochs.

Python programming language and the Keras library were used for training and obtaining the results. With the purpose of evaluating the performance of this architecture, the same metrics of the previous case were used.

### 5.2.3. Performance Analysis of the Three-Tank System Testbed

Indices values obtained for this case are presented in Figures 16 and 17a,b and Table 7. For accuracy, the best scores were obtained when simultaneous attacks happened with values above 0.97. The above is an important result because this situation has been little explored. In terms of recall, class 7 has a slightly fair score, while the other situations have scores above 0.83. Additionally, the F1 score also has high values. The scores show that the proposed architecture allows for high specificity and high sensitivity.

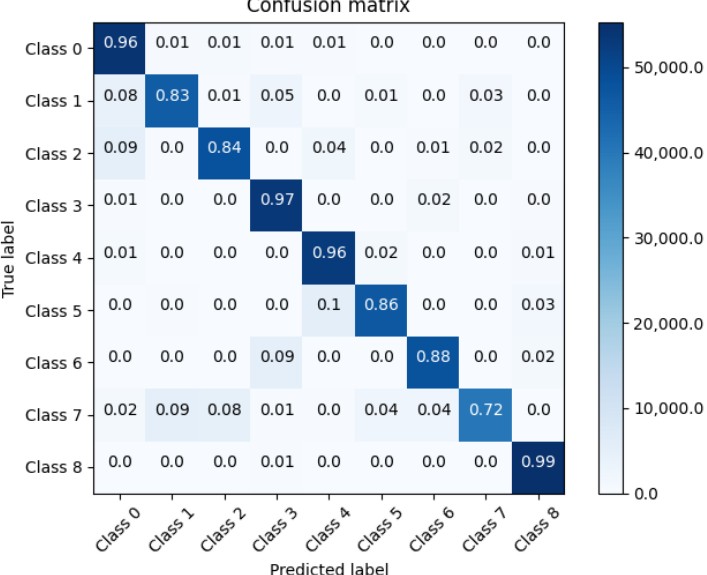

**Figure 16.** Confusion matrix for the three-tank system.

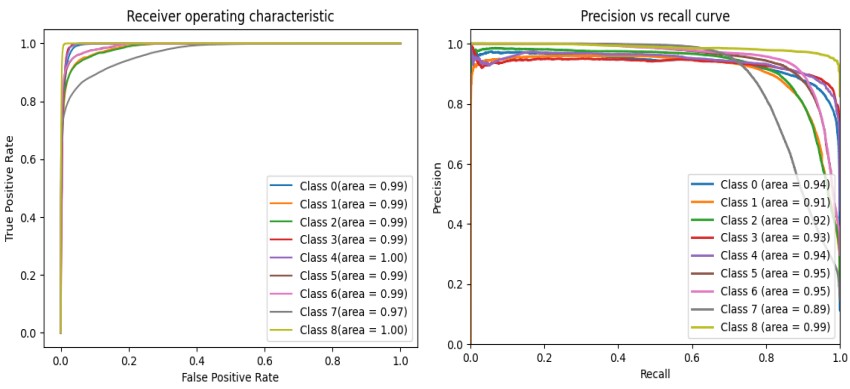

(**a**) ROC Curve.  (**b**) Precision-Recall Curve.

**Figure 17.** ROC and Precision-Recall Curves for Interconnected tank.

**Table 7.** Summary of metrics.

|         | Accuracy | Precision | Recall | F1 Score | TNR  |
| ------- | -------- | --------- | ------ | -------- | ---- |
| Class 0 | 0.97     | 0.83      | 0.96   | 0.89     | 0.98 |
| Class 1 | 0.96     | 0.89      | 0.83   | 0.86     | 0.99 |
| Class 2 | 0.97     | 0.89      | 0.84   | 0.87     | 0.99 |
| Class 3 | 0.98     | 0.86      | 0.97   | 0.91     | 0.98 |
| Class 4 | 0.98     | 0.87      | 0.96   | 0.91     | 0.98 |
| Class 5 | 0.98     | 0.91      | 0.86   | 0.89     | 0.99 |
| Class 6 | 0.98     | 0.92      | 0.88   | 0.90     | 0.99 |
| Class 7 | 0.96     | 0.94      | 0.72   | 0.81     | 0.99 |
| Class 8 | 0.99     | 0.94      | 0.99   | 0.97     | 0.99 |

The alarm indicator was implemented from the classifier in order to know the process state. Since the classifier provides the probability to classify an input data in particular class, the alarm signal is generated taking in to account the maximum value obtained from the classifier. In Figure 18, the alarm indicator is 1 when sensor 1 or 2 is under attack, and 0 when it is not. Additionally, it is discriminated if the attack is DoS or integrity type. The response of the process when it is attacked is shown in Figure 19. Boxes indicate the time instance when the attack occurs in both sensors, according to the alarm signals generated. Red boxes correspond to DoS attacks, and black boxes correspond to integrity attacks.

Additionally, the effect is different, depending on whether it is DoS or integrity attack. The system proposed in this work performed appropriately to detect the occurrence of the cyber attack, as well as the location and type of the attack. As results obtained using convolutional networks were better than those employing RNN or LSTM networks, convolutional networks were then chosen for this proposal.

In summary, the key steps for using the proposed architecture are as follows:

1. Generate an estimated output of the process under a regression model.
2. Generate a residual signal under the comparison of the measured process outputs with estimated outputs.
3. Use a classification model that from some system characteristics, such as control actions, estimated outputs, measured process outputs, and residual signals, allows evaluating if there is an attack in any part of the system.
4. From the detected class, generate alarm signals to report the occurrence of a cyber attack to define the type of attack and the part of the system that is being affected by it.

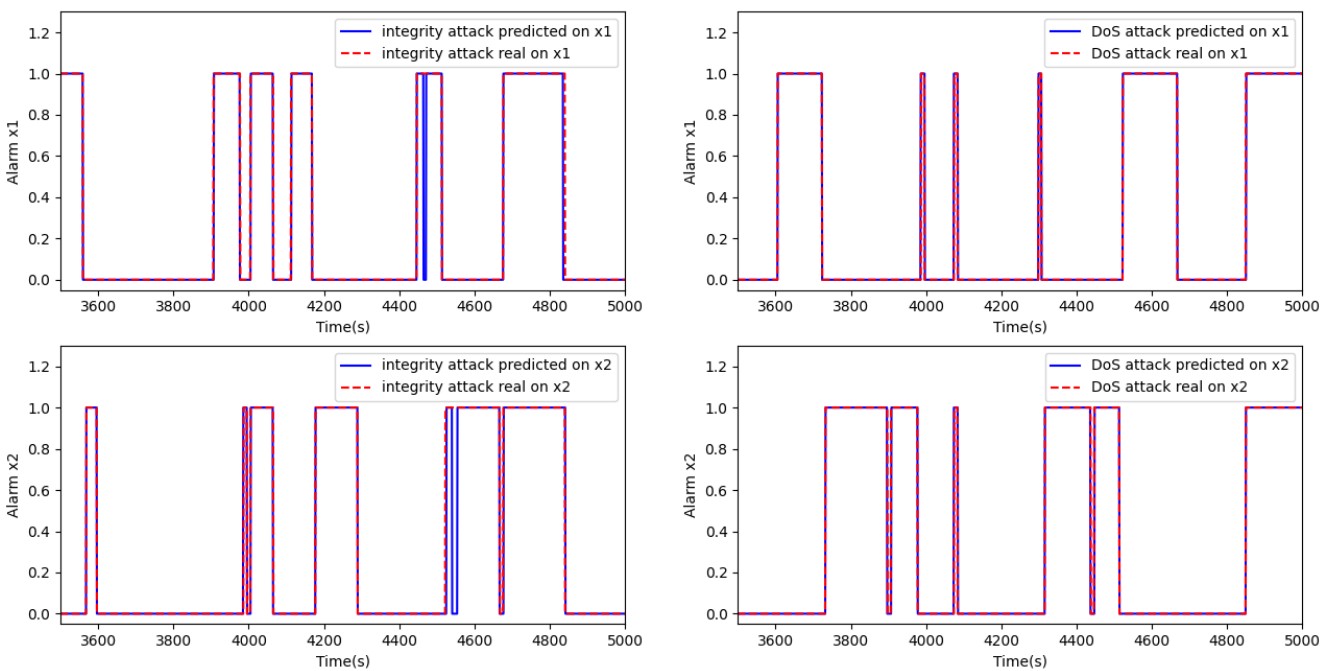

**Figure 18.** Alarm generation.

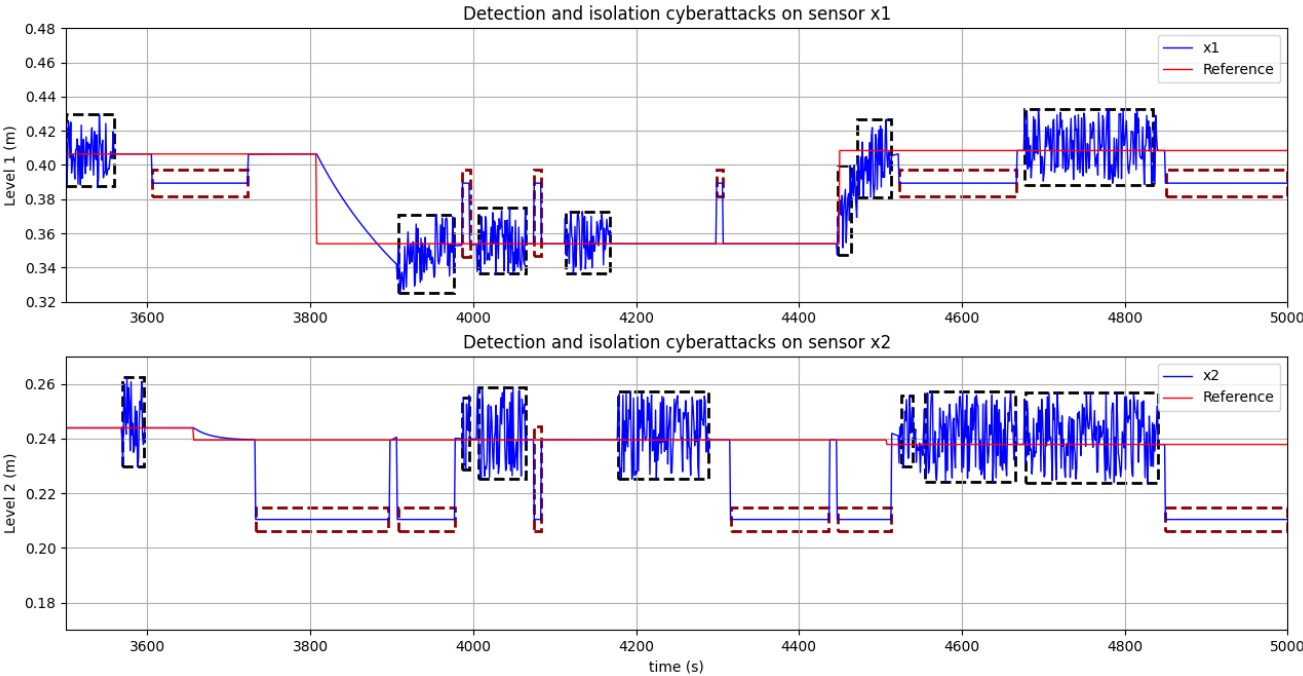

**Figure 19.** Temporal response.

## 6. Conclusions

New applications of industrial automation request great flexibility in the systems, which is supported by the increase in the interconnection between its components. At the same time, it generates a large gap that affects the security of control systems. Current solutions are oriented mainly to avoid the occurrence of attacks, but, regardless, the problems appear; so, recently, the interest in developing new proposals that contribute to detect attacks has grown.

In this work, a new architecture for DoS and integrity cyber attacks detection and isolation in Cyber Physical Systems using one-dimensional Convolutional Neural Networks was presented, thereby overcoming other models that are based on machine learning and

model-based methods, such as the use of Unknown Input Observers. This architecture involves a series of steps to achieve its purpose. The first step was to generate an estimated output of the process under a regression model. The next step was to generate a residual signal under the comparison of the measured process outputs with estimated outputs. Then, a classification model was added whose input data are different characteristics, such as control actions, estimated outputs, measured process outputs, and residual signals. This model allowed for detection and isolation of different eventualities that were defined in classes. Finally, from the detected class, alarm signals were generated that are used to report the occurrence of a cyber attack, allowing to define the type of attack and the part of the system that is being affected by the attack.

The architecture proposed does not use threshold information to detect and isolate attacks, as is the case with model-based methods, such as Unknown Input Observers, which often use this information. These models require an exhaustive selection of these thresholds, which can cause both false detections and anomalous situations that go undetected, and the proposed architecture provides shows advantages over this.

The performance of the proposed architecture was validated by two test benches obtaining satisfactory results compared to other methods. The results on the SWaT dataset allowed observing that, in terms of precision and accuracy, the indexes are very close to the highest scores of other works, and these obtained a score of 0.95. In terms of recall and F1Score metrics, it presented a score of 0.95, which outperforms the previously proposed methods by a good margin. Overall, the proposed system has a high true positive rate and a low false positive rate. On the other hand, the ability of the system to be able to detect and isolate cyber attacks that may occur simultaneously is highlighted, which was presented in the three-tank system testbed. In the defined classes, the accuracy presents scores above 0.96, and the precision is above 0.83, in cases where attacks occur in a single part of the system, while the score is higher than 0.91 in cases where simultaneous attacks occur. In terms of the F1 score metric, the scores are above 0.81, which is a very promising result. Finally, with respect to the recall metric, the scores are above 0.83, in most cases. With the cases presented in this testbed, it was possible to demonstrate the ability of the proposed architecture to detect and locate attacks that may occur simultaneously. This is interesting because these types of experiments are rarely performed, let alone provide evidence of systems that can detect these types of situations, which are not alien to eventualities that may occur in reality. In both cases highlighted, there was a high rate of TNR in each of the classes, ranging between 0.98 and 0.99.

**Author Contributions:** Conceptualization, C.M.P.; methodology, C.M.P. and D.M.-C.; software, C.M.P.; validation, C.M.P. and D.M.-C.; formal analysis, C.M.P.; investigation C.M.P. ; resources, D.M.-C.; data curation, C.M.P.; writing—original draft preparation, C.M.P.; writing—review and editing, C.M.P, D.M.-C., V.I.-J. and A.G.-P.; visualization, D.M.-C. and A.G.-P; supervision, D.M.-C.; project administration, D.M.-C. All authors have read and agreed to the published version of the manuscript.

**Funding:** This research received no external funding.

**Data Availability Statement:** Interested partis can contact the first author about the availability of datasets.

**Conflicts of Interest:** The authors declare no conflict of interest.

## Abbreviations

The following abbreviations have been used in this manuscript:

| | |
|---|---|
| AE | Autoencoder |
| CCE | Categorical crossentropy |
| CPS | Cyber Physical System |
| CNN | Convolutional Neural Network |
| DBN | Deep Belief Networks |
| DoS | Denial of Service |

| FNR | False negative rate |
|---|---|
| FPR | False positive rate |
| GRU | Gated recurrent unit |
| ICS | Industrial Control System |
| IDS | Intrusion Detection System |
| LO | Luenberger Observer |
| LSTM | Long short-term memory |
| MSE | Mean squared error |
| PCA | Principal Component Analysis |
| PLC | Programmable Logic Controller |
| ROC | Receiver Operating Characteristics |
| RNN | Recurrent Neural Network |
| SDA | Stacked denoising autoencoder |
| STAE-AD | Spatio-Temporal Autoencoder for Anomaly Detection |
| SVM | Support Vector machine |
| SWaT | Secure Water Treatment |
| TNR | True negative rate |
| TPR | True positive rate |
| UIO | Unknown Input Observer |

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
