# Peer review of "Detection and Isolation of DoS and Integrity Cyber Attacks in Cyber-Physical Systems with a Neural Network-Based Architecture"

_electronics, doi:10.3390/electronics10182238_

Round 1

Reviewer 1 Report

Reviewer’s Recommendation:

This candidate paper needs major revisions.

Summary

A technique is proposed relevant to detecting DoS and cyber-attacks in systems such as Cyber-physical categories along with an architecture that is based on Neural networks.

General comments

This manuscript is vast work that incorporates deep knowledge in this area of expertise thus various experiments for validation purposes. The authors have worked hard but there are some drawbacks that in my opinion must be faced. It would be a pity such work not to be as perfect as it could be. Some grammatical mistakes exist. Revisions should be done.

Suggested Improvements

Apart from the aforementioned, please address the following needed problems inside the manuscript with different color:

  1. Cyber-attacks pose a significant danger to various installations such as in Attack Detection for Healthcare Monitoring Systems Using Mechanical Learning in Virtual Private Networks over Optical Transport Layer Architecture. Please take it into consideration.
  2. In related works section, you should better mention the utilized databases along with the keywords thus inclusion/exclusion criteria.
  3. Equations (1) and (2) should be simply mentioned in order not a reader to make read also other papers for understanding only two equations. The goal is to attract multidisciplinary works and readers. The diffusion will then rise.
  4. "In order to evaluate the performance of the architecture proposed two test benches were used...". Please revise the expression. It is not shaped correctly.
  5. Figure 4 should be of higher analysis and thus all other figures.
  6. "Attackers changes L_T7 thresholds..." Please revert "changes" to "change". Similarly in ID3 whereas "alters" should be reverted to "alter".
  7. Figure 3 is somehow redundant. It is preferable to mention the facts rather than exhibiting such a diagram whereas the conclusion is obvious. It would be better to embed this diagram if you want to keep it inside Figure 7 with a different color. Then, it could make sense.
  8. Relevant to Precision-Recall analysis which is included in ROC analysis theory please see "Checking for voice disorders without clinical intervention: The Greek and global VHI thresholds for voice disordered patients". It would be nice to draw also a Precision-Recall curve while calculating its cutoff point vs ROC curves cutoff. In general these kind of points are essential. Please read the above publication. It will help.
  9. Relevant to F1 score you should also mention that is equal to 2 x (Precision x Recall)/ (Precision + Recall)
  10. You present ROC curves and not Precision-Recall analysis curves and you present Accuracy, Precision, Recall. When presenting ROC, also sensitivity and specificity need to be calculated and presented.
  11. Before concluding you should summarize your model's steps of attack detection strategy. It will also help others to use your work.

Manuscript Rating:

  1. This candidate paper has much to show. Give some precaution to the needed revisions and the paper will be of high quality.

Author Response

Dear Reviewer, Thanks a lot. We would like to thank to the reviewer for their valuable comments and suggestions that have resulted in the improvement of the completeness and readability of our paper.  We take enough time to make a detailed review of the comments and suggestions and do a complete and careful revision according to the referee's comments and suggestions.  A reply addressing comments and suggestions to the referee is given below to clarify how we addressed the comments and updated the paper.

  1. Cyber-attacks pose a significant danger to various installations such as in Attack Detection for Healthcare Monitoring Systems Using Mechanical Learning in Virtual Private Networks over Optical Transport Layer Architecture. Please take it into consideration.

The aforementioned reference made it possible to update the state of the art and was included in the related works section (rows 178-185).

  1. In related works section, you should better mention the utilized databases along with the keywords thus inclusion/exclusion criteria.

A paragraph was added in the related work section (rows 219-228).

  1. Equations (1) and (2) should be simply mentioned in order not a reader to make read also other papers for understanding only two equations. The goal is to attract multidisciplinary works and readers. The diffusion will then rise.

The paragraph below Equations (1) and (2) was extended to explain the missing terms (rows 253-256).

  1. "In order to evaluate the performance of the architecture proposed two test benches were used...". Please revise the expression. It is not shaped correctly.

The expression was corrected (rows 341-342).

  1. Figure 4 should be of higher analysis and thus all other figures.

A paragraph explaining the Figure was added (rows 357-369).

  1. "Attackers changes L_T7 thresholds..." Please revert "changes" to "change". Similarly in ID3 whereas "alters" should be reverted to "alter".

The expressions were corrected (Table 1).

  1. Figure 3 is somehow redundant. It is preferable to mention the facts rather than exhibiting such a diagram whereas the conclusion is obvious. It would be better to embed this diagram if you want to keep it inside Figure 7 with a different color. Then, it could make sense.

At this point, we had problems understanding it because Figures 3 and 7 are not related. We believe that it referred to Figure 6 instead of Figure 3. If so, we understand the recommendation, but we consider it convenient to keep Figures 6 and 7 so that readers who are not experts in the subject can more easily understand the methodology that was necessary to process the information and improve the results of the algorithms. Only one Figure (Figure 6) containing both histograms was included.

(rows 411-412).

  1. Relevant to Precision-Recall analysis which is included in ROC analysis theory please see "Checking for voice disorders without clinical intervention: The Greek and global VHI thresholds for voice disordered patients". It would be nice to draw also a Precision-Recall curve while calculating its cutoff point vs ROC curves cutoff. In general these kind of points are essential. Please read the above publication. It will help.

The precision-recall curve was added for better analysis (Figure 9 and Figure 17).

  1. Relevant to the F1 score you should also mention that is equal to 2 x (Precision x Recall)/ (Precision + Recall)

Clarification was made is Equation (13) (rows 432-434, 557-559).

  1. You present ROC curves and not Precision-Recall analysis curves and you present Accuracy, Precision, Recall. When presenting ROC, also sensitivity and specificity need to be calculated and presented.

Specificity was added to the summary tables of the metrics. Initially, this metric had not been considered because the other methods to be used for comparison were not established (Table 2 and 6, Figure 9 and 17, rows 447-452, rows 557-558).

  1. Before concluding you should summarize your model's steps of attack detection strategy. It will also help others to use your work.

Added a paragraph in the conclusions (rows 583-592).

Reviewer 2 Report

The paper presents an ML-based architecture for the detection and isolation of the attacks which may occur between the elements of the physical layer and the controller.

The topic is interesting and the paper seems to have some merit. On the other, i can verify that the paper has different problems and there are strong indications that it was written hastily.

Specific comments to improve the paper:

A. The following sentences are indicative of the level of confusion that is created to the reader. Please rephrase them and check all the text.

  1. "IDSs are techniques to prevent vulnerabilities in the devices connected to the system’s network through monitoring data traffic to protect systems".
  2. "These systems consist of three stages"... Obviously you refer to the IDS and not the general concept of the "system". Please rephrase.
  3. "It works with software such as antivirus..."
  4. "Finally, the conclusions are presented" -> Finally, the last section... 

B. The Abstract and the conclusions need to be enhanced with numerical results.

C. At the end of the introduction we need a paragraph which highlights the contributions of the work at hand.

D. In the end of section 2 you would need to justify why the community needs your solution over the rest of the prior art. A comparative table would help to highlight your contributions.

E. Which is the merit of the dates in Table 1? I would propose to keep only the Duration and add the corresponding values there.

Author Response

Dear Reviewer. Thanks a lot. We would like to thank the reviewer for their valuable comments and suggestions that have resulted in the improvement of the completeness and readability of our paper.  We take enough time to make a detailed review of the comments and suggestions and do a complete and careful revision according to the referee's comments and suggestions.  A reply addressing comments and suggestions to the referee is given below to clarify how we addressed the comments and updated the paper.

A. The following sentences are indicative of the level of confusion that is created to the reader. Please rephrase them and check all the text.

  1. "IDSs are techniques to prevent vulnerabilities in the devices connected to the system’s network through monitoring data traffic to protect systems".

The sentence was corrected (rows 58-59)

  1. "These systems consist of three stages"... Obviously you refer to the IDS and not the general concept of the "system". Please rephrase.

The sentence was corrected (row 59)

  1. "It works with software such as antivirus..."

The sentence was corrected (rows 69-74)

  1. "Finally, the conclusions are presented" -> Finally, the last section... 

The sentence was corrected (row 112).

B. The Abstract and the conclusions need to be enhanced with numerical results.

The numerical results in the two sections (rows 18-24 and rows 598-618) were added.

C. At the end of the introduction we need a paragraph  which highlights the contributions of the work at hand.

A paragraph was added at the end of the introduction (rows 96-108).

D. In the end of section 2 you would need to justify why the community needs your solution over the rest of the prior art. A comparative table would help to highlight your contributions.

A paragraph was added at the end of the section (rows 229-251).

E. Which is the merit of the dates in Table 1? I would propose to keep only the Duration and add the corresponding values there.

The content of Table 1 was changed by eliminating the date columns and the labeled times.

Reviewer 3 Report

Please find the attached.

Author Response

Dear Reviewer, Thanks a lot. We would like to thank to the reviewer for their valuable comments and suggestions that have resulted in the improvement of the completeness and readability of our paper. We take enough time to make a detailed review of the comments and suggestions and do a complete and careful revision according to the referee's comments and suggestions. 
A reply addressing comments and suggestions to the referee is given below to clarify how we addressed the comments and updated the paper.

  1. (page 5, Section 3): Although the section is “Problem statement”, the reviewer had a hard time finding a properform ofthe problemthe authors wanted to solvein Section 3. Would it be possible to more clearly state what to solve?

Added paragraphs in Section 3 (rows 252-253 and rows 264-270).

  1. (page 10–11, Fig. 6 and 7): The plot does not have labels on the axes.

The figures have been corrected, now it is Figure 6 (row 412).

  1. (page 13, Table 3): The methods have autoencoders (reference [69]) as compared schemes. However,since this paper mainly focuses on classifications, autoencoders may not be adequate compared architectures. The reviewer would like to see how autoencoders are utilizedfordetectingand isolating cyberattacks.

Architectures based on autoencoders seek to reconstruct the time series of printed data. In this case, these estimated data are compared with the real data obtained from the data set, and a residual similar to that presented in Equation 5 is generated. And later it is evaluated in a similar way to Equation 6 to detect the occurrence of the cyber attack

The idea that is currently had with autoencoders is to be able to have a representation of the states of the system with the characteristic that these architectures have, where the training process is carried out in an unsupervised way.

  1. (page 19, Fig. 16 and 17): The training loss and validation loss curves are not distinguishable.

The figures were corrected, now it is Figure 15 (row 547).

Reviewer 4 Report

The article proposes an architecture based on artificial neural networks for detection and isolation of cyberattacks Denial of Service (DoS) and integrity in the Cyber-Physical Systems (CPS), aiming to create a methodology to where the thresholds level in data for alarm are obtain from data itself. Being more precise an architecture for DoS and integrity cyber attacks detection and isolation in Cyber Physical Systems using Convolutional Neural Networks 1-dimensional was presented, where does not use threshold information to detect and isolate attacks, but learn them from historical (“normal”) and simulation attacks (“anomalies”) data.

The problem of ensuring security against cyber-attacks and recognizing the correct and incorrect operation of systems is crucial for the practice, as the costs of both false-positive and false-negative decisions are high. Hence, the article is important for practice and tries to connect new possibilities by using AI for solving classic Cyber ​​Physical Systems control (steering) problems.

As new applications of industrial automation request great flexibility in the systems which is supported by the increase in the interconnection between its components, Cyber ​​Physical Systems (CPSs) are interconnected using communication networks to its physical layer, where the physical infrastructure of the system, sensors, and actuators are located, by the network layer with aa cybernetic layer (steering) and are vulnerable to cyber attacks. Their detection and distinguishing from the “almost normal” operation of the system is not easy and the methodologies are usually based on the indication - for each sensor and set of sensors - of the appropriate level of value as an alarm one (thresholds).

The Authors propose to introduce an algorithm which, based on historical data on normal sensor indications and data with “test” (simulated) anomalies, will independently recognize “the alarm” indication of set of sensors. Such a solution is reasonable and is based on the potential contained in the capabilities of neural networks, but is largely dependent on the learning processes.

The reviewer lacked clear guidelines in the text as to the preparation of these learning processes, i.e. criteria to indicate on which examples of anomalies it is worth teaching these anomaly recognition systems, and how to recognize a sufficient range of anomalies that were prepared for the learning system. In one of the applications, the Authors pointed that "The integrity attacks were implemented by changing the modified variable in a range of 5% to 8% of its measured value", without discussing in more detail the criteria for recognizing such a scale of indicator deviations or which part of the indicators set, as well reflecting on possible symptoms of different classes of anomalies.

The strong part of the article is testing the proposed solution on some real data. As emphasized by the Authors "The performance of the architecture proposed was validated by two test benches obtaining satisfactory results compared to other methods", while showing the system was able to detect and isolate cyber attacks that can occur simultaneously, which is not always easy for alternative methods (based on the given thresholds).

The reviewer was pleased to read the presentation of the theoretical field (Related works), although he lacked a clear specification of the reasons why anti-attacks systems adequate for office applications are not adequate for industrial applications, and which, according to the Authors, are necessary to be considered for control in industrial systems.

Author Response

Dear Reviewer, Thanks a lot. We would like to thank to the reviewer for their valuable comments and suggestions that have resulted in the improvement of the completeness and readability of our paper. 
We take enough time to make a detailed review of the comments and suggestions and do a complete and careful revision according to the referee's comments and suggestions. A reply addressing comments and suggestions to the referee is given below to clarify how we addressed the comments and updated the paper. 

  1. The reviewer lacked clear guidelines in the text as to the preparation of these learning processes, i.e. criteria to indicate on which examples of anomalies it is worth teaching these anomaly recognition systems, and how to recognize a sufficient range of anomalies that were prepared for the learning system. In one of the applications, the Authors pointed that "The integrity attacks were implemented by changing the modified variable in a range of 5% to 8% of its measured value", without discussing in more detail the criteria for recognizing such a scale of indicator deviations or which part of the indicators set, as well reflecting on possible symptoms of different classes of anomalies.

A paragraph was added at the end of the related work section to clarify the types of anomalies to be addressed (rows 229-251). Additionally, an explanation of the reason for the range of values selected was added (row 492 and rows 498-500).

  1. The reviewer was pleased to read the presentation of the theoretical field (Related works), although he lacked a clear specification of the reasons why anti-attacks systems adequate for office applications are not adequate for industrial applications, and which, according to the Authors, are necessary to be considered for control in industrial systems.

A paragraph was added at the beginning of the related work to clarify this doubt (rows 121-131).

Round 2

Reviewer 1 Report

The authors have conducted considerable corrections and revisions. Still, there are a few minor revisions that should be accomplished:

  1. In related works section, you should better mention the utilized databases along with the keywords thus inclusion/exclusion criteria – This comment has not been addressed.
  2. Before concluding you should summarize your model's steps of attack detection strategy. It will also help others to use your work. – E.g. this should include the key steps in bullet-style but before the conclusion.

Sincerely,

The reviewer

Author Response

We take enough time to make a detailed review of the comments and suggestions and do a complete and careful revision according to the referee's comments and suggestions. A reply addressing comments and suggestions to the referee is given below to clarify how we addressed the comments and updated the paper.

Dear Reviewer 1, Thanks a lot.

  1. In related works section, you should better mention the utilized databases along with the keywords thus inclusion/exclusion criteria – This comment has not been addressed.

 A paragraph has been added in Section 2 (rows 207-215)

  1. Before concluding you should summarize your model's steps of attack detection strategy. It will also help others to use your work. – E.g. this should include the key steps in bullet-style but before the conclusion.

 A paragraph has been added before the conclusions (rows 564-573).

Reviewer 2 Report

It seems that the authors have improved their work in this version. Still, they have not addressed my comment to add a new table to the related work section, so that they could complete the presentation part.

  1. Please correct the following sentence in the introduction "Finally, the last section the conclusions are presented".
  2. "...attacks are of a different nature" -> please explain
  3. I think that the abbreviations should be moved at the end, probably in an appendix.

Author Response

We take enough time to make a detailed review of the comments and suggestions and do a complete and careful revision according to the referee's comments and suggestions. A reply addressing comments and suggestions to the referee is given below to clarify how we addressed the comments and updated the paper.

Dear Reviewer 2, Thanks a lot.

It seems that the authors have improved their work in this version. Still, they have not addressed my comment to add a new table to the related work section, so that they could complete the presentation part.

Table 1 was added, where a comparison of the different works is made presenting their advantages and disadvantages, thus identifying the direction and contribution of our work. (rows 225-227).

  1. Please correct the following sentence in the introduction "Finally, the last section the conclusions are presented".

Corrected sentence (rows 105-106).

  1. "...attacks are of a different nature" -> please explain

Expression changed (row 101)

In that part we mean that attacks can be of different types.

  1. I think that the abbreviations should be moved at the end, probably in an appendix.

Abbreviations were moved to the end (rows 618-622).